# Effectiveness of shared decision-making for glycaemic control among type 2 diabetes mellitus adult patients: A systematic review and meta-analysis

Edosa Tesfaye Geta[1]*, Dufera Rikitu Terefa[1], Wase Benti Hailu[1], Wolkite Olani[1], Emiru Merdassa[1], Markos Dessalegn[1], Miesa Gelchu[2], Dereje Chala Diriba[3]

1 School of Public Health, Institute of Health Sciences, Wollega University, Nekemte, Ethiopia, 2 School of Public Health, Institute of Health, Bule Hora University, Bule Hora, Ethiopia, 3 School of Nursing and Midwifery, Institute of Health Sciences, Wollega University, Nekemte, Ethiopia

* edotesfa@yahoo.com

**Data Availability Statement:** All relevant data are within the manuscript and its Supporting information files.

## Abstract

### Background

In diabetes care and management guidelines, shared decision-making (SDM) implementation is explicitly recommended to help patients and health care providers to make informed shared decisions that enable informed choices and the selection of treatments. Despite widespread calls for SDM to be embedded in health care, there is little evidence to support SDM in the management and care of diabetes. It is still not commonly utilized in routine care settings because its effects remain poorly understood. Hence, the current systematic review and meta-analysis aimed to evaluate the effectiveness of SDM for glycaemic control among type 2 diabetes adult patients.

### Methods

Literature sources were searched in MEDLINE, PubMed, Cochrane library and HINARI bibliographic databases and Google Scholar. When these records were searched and reviewed, the PICO criteria (P: population, I: intervention, C: comparator, and O: outcome) were applied. The extracted data was exported to RevMan software version 5.4 and STATA 17 for further analysis. The mean differences (MD) of glycated hemoglobin (HbA1c) were pooled using a random effect model (REM), and sub-group analysis were performed to evaluate the effect size differences across the duration of the follow-up period, modes of intervention, and baseline glycated hemoglobin level of patient groups. The sensitivity analysis was performed using a leave-one-out meta-analysis to quantify the impact of each study on the overall effect size in mean difference HbA1c%. Finally, the statistically significant MD of HbA1c% between the intervention groups engaged in SDM and control groups received usual care was declared at $P < 0.05$, using a 95% confidence interval (CI).

**Funding:** The author(s) received no specific funding for this work.

**Competing interests:** The authors have declared that no competing interests exist.

## Results

In the database search, 425 records were retrieved, with only 17 RCT studies fulfilling the inclusion criteria and were included in the meta-analysis. A total of 5416 subjects were included, out of which 2782(51.4%) were included in trial arms receiving SDM and 2634 (48.6%) were included in usual diabetes care. The Higgins ($I^2$) test statistics were calculated to be 59.1%, P = 0.002, indicating statistically significant heterogeneity was observed among the included studies, and REM was used as a remedial to estimate the pooled MD of HbA1c% level between patients who participated in SDM and received usual care. As a result, the pooled MD showed that the SDM significantly lowered HbA1c by 0.14% compared to the usual care (95% CI = [-0.26, -0.02], P = 0.02). SDM significantly decreased the level of HbA1c by 0.14% (95% CI = -0.28, -0.01, P = 0.00) when shared decisions were made in person or face-to-face at the point of care, but there was no statistically significant reduction in HbA1c levels when patients were engaged in online SDM. In patients with poorly controlled glycaemic level ($\geq$ 8%), SDM significantly reduced level of HbA1c by 0.13%, 95% CI = [-0.29, -0.03], P = 0.00. However, significant reduction in HbA1c was not observed in patients with < 8%, HbA1c baseline level.

## Conclusions

Overall, statistically significant reduction of glycated hemoglobin level was observed among T2DM adult patients who participated in shared decision-making compared to those patients who received diabetes usual care that could lead to improved long-term health outcomes, reducing the risk of diabetes-related complications. Therefore, we strongly suggest that health care providers and policy-makers should integrate SDM into diabetes health care and management, and further study should focus on the level of patients' empowerment, health literacy, and standardization of decision supporting tools to evaluate the effectiveness of SDM in diabetes patients.

## Introduction

Diabetes mellitus (DM) is a group of metabolic diseases in which the person has high blood glucose (blood sugar), either because of insulin production is inadequate (type 1 diabetes mellitus), or because of the body's cells do not respond properly to insulin (type 2 diabetes mellitus (T2DM), or both. In T2DM, the body does not produce enough insulin for proper function, or the cells in the body do not react to insulin (insulin resistance), and approximately 90% of all cases of diabetes worldwide are T2DM [1].

About 462 million people worldwide (4.4% of those in the 15–49 age group, 15% of those in the 50–69 age group, and 22% of those over 70) had type 2 diabetes in 2017. This represents 6.28% of the global population and a prevalence rate of 6059 cases per 100,000 people. Diabetes is the tenth most common cause of death, accounting for over a million fatalities annually. Worldwide, particularly in developed regions, the prevalence of diabetes mellitus is increasing at a significantly faster rate [2].

The glycated hemoglobin (HbA1c) test, continuous glucose monitoring (CGM), and self-monitoring of blood glucose (SMBG) are methods used to evaluate glucose control. The metric used in clinical trials so far to show the advantages of better glycaemic control is HbA1c. The

main method for evaluating glycaemic control is the HbA1c test, which has a good predictive ability for complications from diabetes [3]. In general, glycated hemoglobin (HbA1c) ≤7% (53 mmol/mol) is considered a favorable indicator of glycaemic control and is the recommended treatment goal [4].

Many practice recommendations for diabetes management adopted an intensive, glucose-centric approach based on findings from glycaemic control studies, consistently using HbA1c as a metric of metabolic control and effectiveness. Shared decision-making (SDM), incorporating patients' needs, values, and preferences with evidence-based treatment, and least disruptive diabetic care strategies are all components of a person-centered approach to diabetes management. In addition, it provides both healthcare providers and patients with helpful guidance on how to provide this type of treatment [5]. Improving glycaemic control demands the participation of patients in decisions about how to manage the condition and accommodate to the facts about self-care; this is the reason why SDM enhances glycaemic control [6].

According to diabetes care and management guidelines, SDM implementation is explicitly recommended to help patients and health care providers to make informed shared decisions that enable informed choices and the selection of treatment that best fits individual patient needs, values, and preferences [7].

Patient decision aids are just one component of a tool set that healthcare providers can use to promote SDM, which is a process that requires a collaborative partnership between patients and healthcare providers [8] and it is a process that can be used in any clinical action, whether it is therapeutic, preventive, or diagnostic, in the doctor-patient interaction [9].

Through SDM and mutually beneficial conversations, patient decision aids could offer a straightforward and user-friendly approach, potentially increasing diabetes health literacy. Building a strong rapport during the clinical encounter is important for achieving SDM because it facilitates the sharing of information and allows patients to consider and communicate their preferences and opinions when making decisions. SDM is commonly acknowledged as a fundamental strategy for enhancing patient-centered care. It is still not commonly utilized in routine care settings, and its effects remain poorly understood. Individuals with diabetes frequently have complex lives and co-existing with other chronic medical conditions. The optimal management choice for a patient depends on considering his or her individual psychological, social, and biological circumstances what matters to him or her, why each of the choices are important to him or her, and how each option compares in terms of advantages, disadvantages, costs, and challenges [8–17].

Patient-centered diabetes care requires SDM which is prompted by decision aids. Decision aids improve decisional outcomes without significant effect on clinical outcomes. It was designed for point-of-care use with type 2 diabetes patients promoted shared decision making in non-academic and rural primary care practices, but their efficacy is unclear [18].

There has been a concern that medical SDM has not sufficiently incorporated the individual circumstances and concerns of patients because the SDM tools used during the consultation can trigger cost conversations but are insufficient to support them and do not improve clinical outcomes [19]. In contrast to this, the study conducted in the USA indicated that an SDM intervention for individuals with type 2 diabetes appears to be feasible, and outcomes in primary care and HbA1c levels showed a trend towards improvement [20].

The SDM, an innovative aid that effectively involved patients with T2DM in decision-making significantly improved the knowledge and involvement of patients in decision-making. Utilization of decision aids in SDM is effective to reduce decisional conflicts and improve knowledge on the disease and treatment options, awareness of risk, and satisfaction with decision made, but did not improve their adherence to medication and HbA1c [9]. The evidence

indicates an association between SDM and enhanced patient perceptions of risk, patient knowledge, and decision quality.

There is scant evidence associating SDM with medication adherence, quality of life, patient satisfaction, glycaemic control, or physician trust. In the previous study [21], further study was recommended to gain insight from the evidence, which clarifies the possible clinical importance of SDM interventions in the management of type 2 diabetes. Despite widespread calls for shared decision-making to be embedded in health care, there is little evidence to inform shared decision-making in diabetes management and care by encouraging SDM because of the uncertainty of the effectiveness of SDM for glycaemic control, which is the primary diabetes treatment goal. Hence, the current systematic review and meta-analysis aimed to evaluate the effectiveness of shared decision-making compared to usual diabetes care for glycaemic control among type 2 diabetes mellitus adult patients.

## Methods

### Data sources and searches

The protocol of the review was registered on PROSPERO with registration ID: CRD42024498413, available at https://www.crd.york.ac.uk/prospero/display_record.php?ID=CRD42024498413. In the protocol, the research question, research objectives and the PICO criteria used were included.

Literature sources were searched in bibliographic data bases: MEDLINE, PubMed, Cochrane library, HINARI and Google Scholar using search terms for shared decision-making, type 2 diabetes and glycated hemoglobin using the search strategy; shared decision making OR SDM OR patient participation OR patient centered OR patient involvement OR patient engagement AND glycaemic control OR glycated hemoglobin level OR HbA1c level AND diabetes OR diabetes mellitus type 2 OR T2DM (S1) in January 2024. All relevant studies published before December 31, 2023, were searched. The PRISMA 2020 checklist [22] was used for the guidance of this review in the design, conduct, and reporting of the review (Fig 1).

### Study selections and eligibility

All retrieved records were evaluated according to their titles and abstracts that were specified as eligibility criteria in the review protocol. When these records were searched and reviewed, the PICO criteria were applied considering population (P): all adult patients (above 18 years of age) clinically diagnosed with T2DM, intervention (I): shared decision-making (SDM) between T2DM patients and care providers, comparator(s)/control (C): usual care of T2DM management and care, and outcome (O): the primary outcome of the current review was the level of HbA1c% among T2DM patients since the primary goal of the T2DM management and care is to control HbA1c level.

There was no restriction on the studies by location, where they were conducted, or year of publication. In addition, all randomized trials were included in the current review. However, the studies published in other languages other than English were excluded from the review. Articles with unclear methodologies, unavailable full-text papers, and articles that did not indicate the outcome of interest were excluded.

All the articles searched from databases were imported into EndNote version 21, and duplicates were checked and removed. Accordingly, 425 records were identified, 27 records were removed because of duplication, 398 records were screened for titles and abstracts, 330 records were excluded by two reviewers (ETG and DRT) following eligibility criteria after titles and abstracts were screened, 51 full articles were excluded with reasons, and 17 articles were assessed for eligibility and included in quantitative synthesis (meta-analysis).

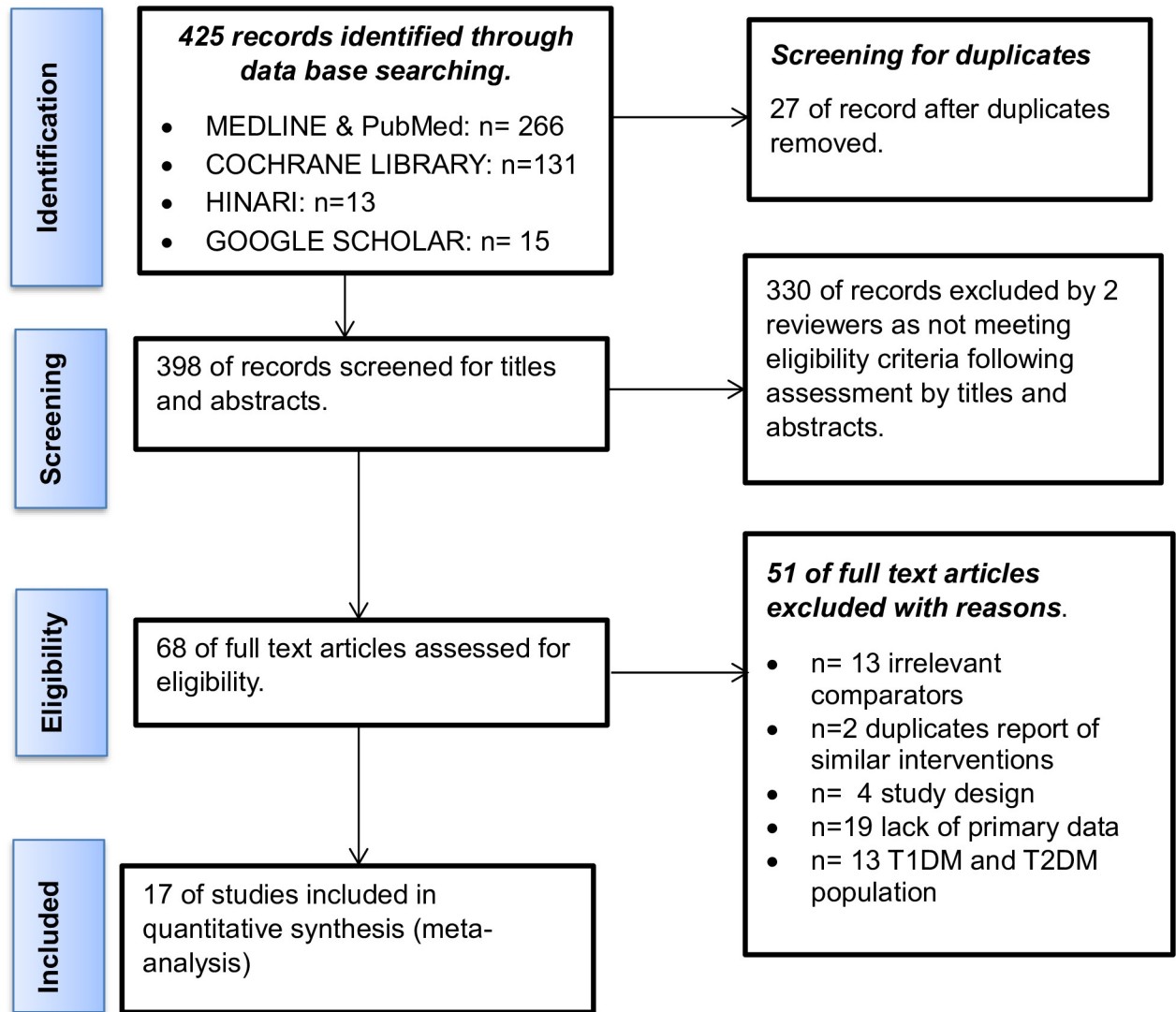

**Fig 1. PRISMA for a systematic review and meta-analysis flow chart for effectiveness of shared decision-making for glycaemic control among type 2 diabetes mellitus adult patients, 2024.**

## Data extraction and quality assessment

Based on the predefined inclusion criteria, two authors (EM and WO) independently assessed and identified articles by their titles, abstracts, and full texts. The screened articles were compiled, and when any disagreements occurred, they were rechecked and handled by inviting and discussing them with the third author (WBH). The standardized data categories drawn from the Cochrane review were used for data extraction [23] and, using the tool, the data were extracted on each study's design, setting, population/participants, sample size, interventions/exposure, control, and outcomes (S2).

In order to assess the quality of the studies included in quantitative synthesis (meta-analysis), a Cochrane risk of bias (RoB 2) assessment checklist was used, having six domains: random sequence generation and allocation concealment (selection bias), blinding of participants and persons who participated in intervention (performance bias), blinding of outcome

assessment (detection bias), incomplete outcome data (attrition bias), selective reporting (reporting bias), and other biases. The studies were judged on each dimension as low risk (green), unclear risk (yellow), and high risk (red) and presented as Cochrane-style risk of bias figures 'traffic light plot' and summarized in Fig 2a and 2b.

The publication bias was visually inspected from funnel plots for the effectiveness of SDM compared to usual care to control glycated hemoglobin for meta-analysis. As a result, the mean of HbA1c% contains some points scattered showing asymmetry, and robustly the publication bias was detected (Fig 3).

### Strategy for data synthesis

The articles that were included in the review were exported to RevMan 5.4 and STATA 17 software using the first author's name and year of publication for analysis, and the results of the review were presented using forest plots. The heterogeneity of the studies was evaluated using Higgins ($I^2$) to identify the heterogeneity problem. The $I^2$ test statistics were used to calculate the percentage of total variance due to heterogeneity across the studies. Following Cochrane recommendation [24], the observed statistical heterogeneity was assessed with $Chi^2$ = 38.34, df = 16, (P = 0.00), a p < 0.1 indicates statistically significant heterogeneity) and quantified by using $I^2$ = 51. 9%, with $I^2$ = 30% to 60% represents moderate heterogeneity. However, the heterogeneity was controlled using a random-effects model (REM) for the meta-analysis as a remedial measure to estimate the effects of SDM in terms of mean differences in HbA1c%.

Moreover, the subgroup analysis by mode of intervention (in-person vs. online: face-to-face at point of care vs. web-based or telephone-based, duration of intervention follow-up, and baseline HbA1c level) were performed to identify the possible source of heterogeneity, and the mean differences across subgroups were assessed using the random-effects model with a 95% CI. Finally, a leave-one-out sensitivity meta-analysis was performed to assess the robustness of the findings.

## Results

### Study selection and characteristics

In data base search, 425 recorded were retrieved with only 17 studies [25–41] fulfilled the inclusion criteria and included in meta-analysis. All the articles included in the current review; 17 of them were randomized controlled trials (RCTs). Out of 17 studies; 6(35.3%) of them conducted in USA [30, 34, 37, 38, 40, 41], 4(23.5%) in Germany [25, 26, 36, 39], 2(11.7%) in UK [29, 31], 1(5.9%) in Greece [28], 1(5.9%) in Dutch [27], 1(5.9%) in Ireland [32], 1(5.9%) in Nepal [33], 1(5.9%) in Netherland and Spain [35] and the studies were published in between 1991 and 2023 (Table 1).

### Quality of the studies

From a total 17 studies included in the current review, 8 studies [25, 26, 29, 32, 33, 35, 39, 40] were judged as having low risk of bias in all dimensions of RoB assessment, 2 studies [27, 41] having high risk of bias in two dimensions, 2 studies [28, 30] having high risk of bias in one dimension, 5 studies [27, 28, 31, 35, 38] having unclear risk of bias in one dimension and 2 studies [30, 36] having unclear risk of bias in two dimensions and only 1 study [37] having unclear risk of bias in three dimensions (Fig 2a and 2b).

(a)

(b)

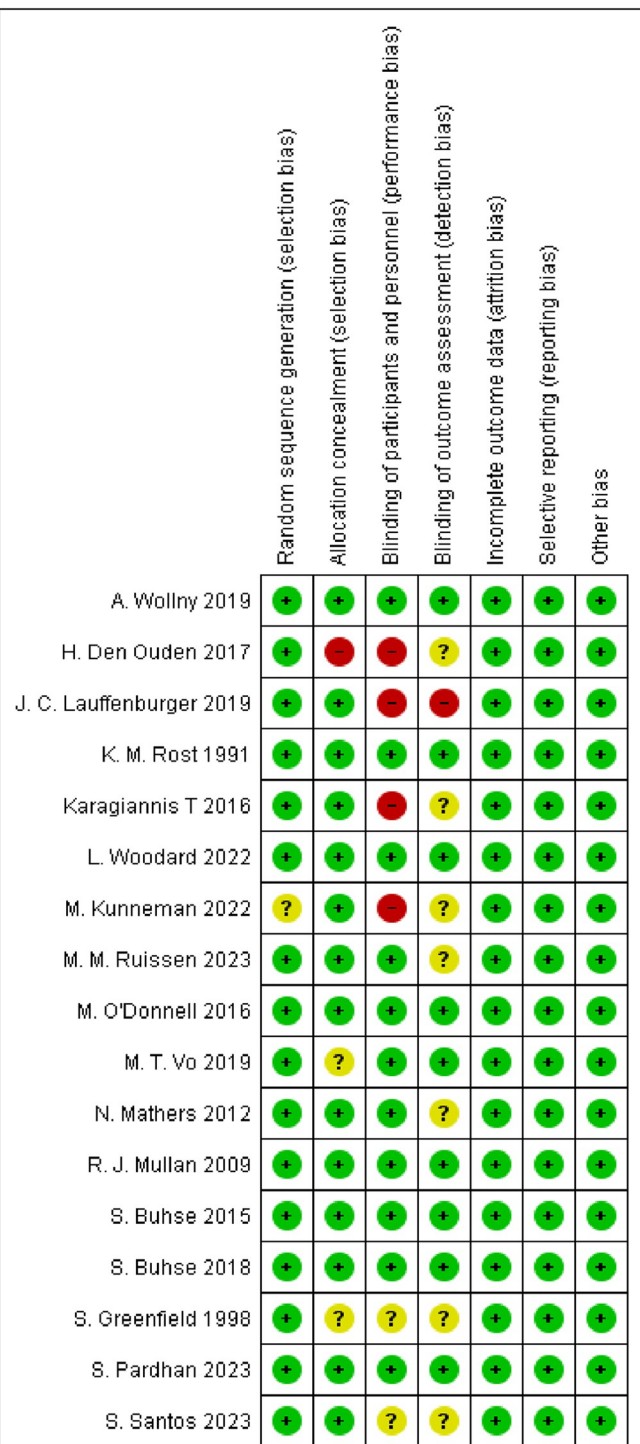

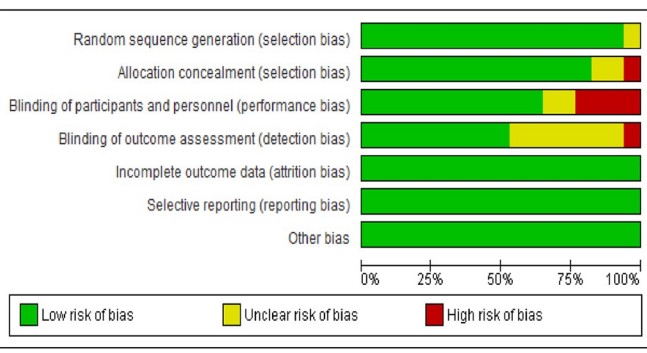

**Fig 2. a.** Risk of bias summary: review authors' judgements about each risk of bias item for each included study. **b.** Risk of bias graph: review authors' judgements about each risk of bias item presented as percentages across all included studies.

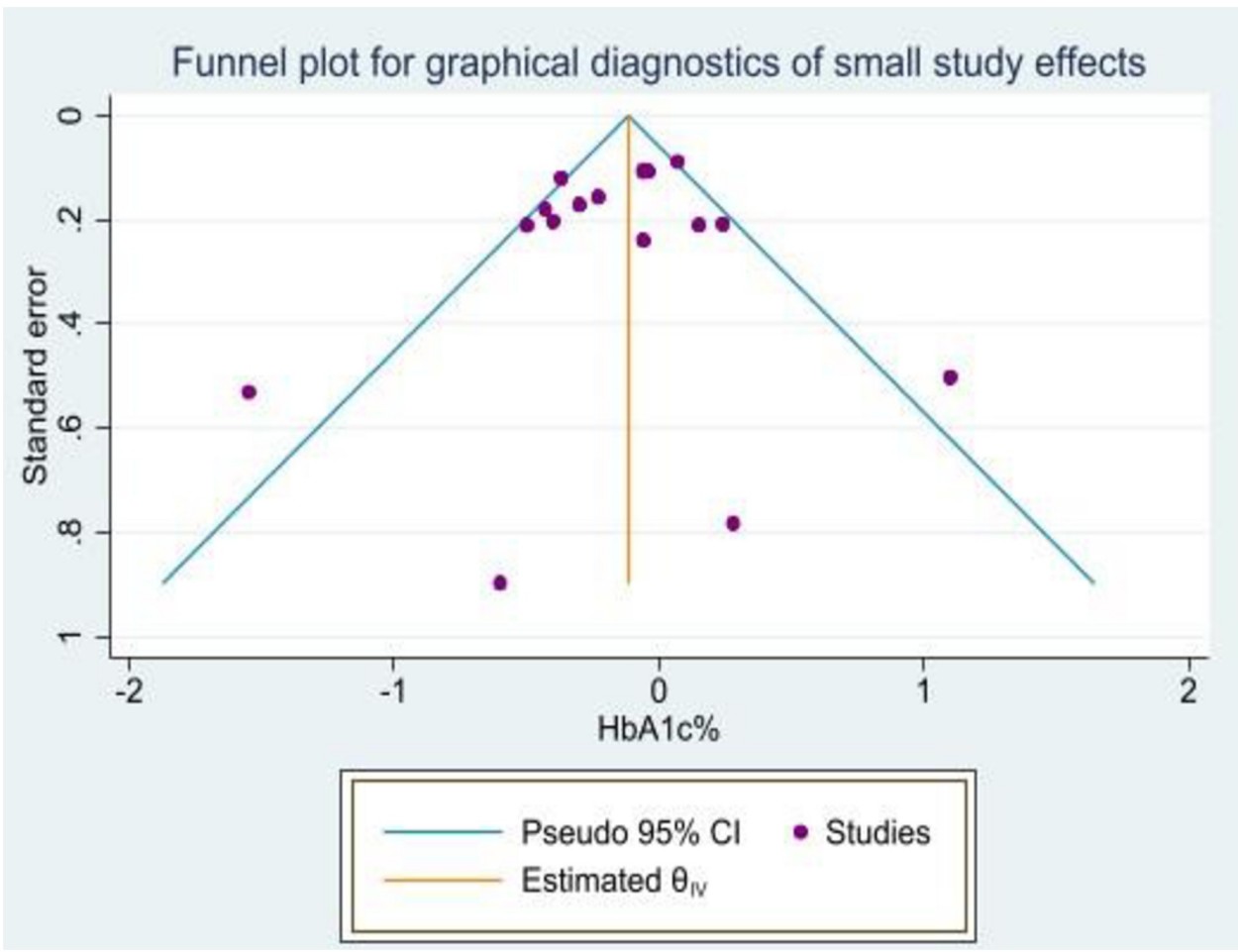

**Fig 3. Funnel plot of comparison: Shared decision-making vs. Usual care effectiveness for glycaemic control in T2DM adult patients, 2024.**
This detected publication bias has downgraded the overall level of certainty of the evidence generated by the current meta-analysis.

## Investigation of publication bias

The publication bias was visually inspected from funnel plots for the effectiveness of SDM compared to usual care to control glycated hemoglobin levels for meta-analysis. As a result, the mean of HbA1c% contains some points scattered, showing asymmetry, and robustly, the publication bias was detected (Fig 3). This detected publication bias has downgraded the overall level of certainty of the evidence generated by the current meta-analysis and the overall certainty of the evidence was graded as moderate.

## Participants characteristics

A total of 5416 subjects were included, out of which 2782(51.4%) were included trial arms receiving shared decision making as intervention and 2634(48.6%) of them included in control arm receiving usual care. The median of follow up period was 12 months. Accordingly, the follow-up duration of nine studies (52.9%) [25, 26, 28, 29, 32–35, 37] was < 12 months whereas the rest of the studies were followed for ≥12 months [27, 30, 31, 36, 38–41]. The mode of delivery of the intervention, in 15 studies [25–34, 36–40] it was face to face (in- person) during the

**Table 1. Characteristics of the studies included in systematic and meta-analysis for the effectiveness of shared decision-making for glycaemic control among type 2 diabetes mellitus adult patients, 2024.**

| Study | Study population | Study settings | Interventions | Mode of intervention (SDM) delivery | HbA1 level (%) baseline, mean (SD) | Follow-up period in month | Main conclusions |
|---|---|---|---|---|---|---|---|
| S.Buhse, et al. 2018 [25] | T2DM patients of age group 40–69 years (N = 279), | The trial was conducted in single-center diabetes clinic providing care according to the national disease management programme in Germany. | ISDM-P comprises a patient decision aid; A single session targets a group of 4–6 patients for 90 minutes, educational elements were illustrating wall charts and worksheets, language summary and designed a leaflet and provider training comprised a training DVD. | Face to face (in-person) at point of care. | I = 6.9 (0.7) C = 7.2 (0.7) | 6 | Despite no significant HbA1c difference was not observed between the groups, patients in the SDM group achieved their HbA1c goals than patients in control group. |
| S. Santos, et al. 2023 [36] | Adult patients with poorly controlled T2DM on improving blood glucose levels (N = 547). | DEBATE trial was conducted at diabetic clinic patient-centered communication and use of an electronic decision aid in Germany. | DEBATE trial for shared goal setting within the process of SDM; educational peer visit and on-site visit, oral presentation, introduction of the decision-aid tool and discussion. | Face to face patient-centered communication at point of care. | I = 9.00 (0.97) C = 9.22 (1.10) | 24 | Shared goal setting with T2DM patients targeting on HbA1c-levels had no significant impact on goal achievement. It may be assumed, that shared goal setting on patient-related clinical outcomes within the process of SDM has not been fully captured yet. |
| S. Pardhan, et al. 2023[33] | Above 18 years of age with newly diagnosed T2DM patients (N = 242). | Trial was conducted in newly diagnosed T2DM patients with FBS ≥126mg/dL in Nepal. | Multidisciplinary diabetic education program that constituted short video clips in the Nepali language which was individualized patient education to empower patients for SDM. | Face to face individualized patient education at point of care | I = 6.1 (1) C = 6.6 (1.2) | 3 | The HbA1c level was found to decrease from the baseline to three months follow-up visit significantly among patients in intervention compared to among patients in control group. |
| M. M. Ruissen, et al. 2023 [35] | Patients with T2DM aged >18 years (N = 118). | POWER2DM trial was conducted in patients with T2DM from hospital outpatient diabetes clinics in the Netherlands and Spain. | Web-based and mobile POWER2DM SMSS-individualized patient education) to support patients and healthcare professionals in shared decision-making | Online (Web-based, telephone based). | I = 7.80 (1.3) C = 7.7 (1.3) | 9.5 | POWER2DM improved HbA1c levels among patients in intervention group compared to among patients in control group with usual care. |
| L. Woodard, etal. 2022 [40] | Adults with uncontrolled T2DM. (N = 280) | Uncontrolled T2DM (HbA1c level >8.0%) in Veterans Affairs clinics across Illinois, Indiana, and Texas, USA. | Empowering Patients in Chronic Care (EPICC)-based on a collaborative goal-setting led by health care professionals. | Face to face group patient education at point-of-care. | I = 9.11 (1.60) C = 9.06 (1.32) | 6 | EPICC using patient-driven goal setting and motivational interviewing delivered by usual care clinicians lowered HbA1c among patients in intervention group compared to patients in control group. |

*(Continued)*

**Table 1.** (Continued)

| Study | Study population | Study settings | Interventions | Mode of intervention (SDM) delivery | HbA1 level (%) baseline, mean (SD) | Follow-up period in month | Main conclusions |
|---|---|---|---|---|---|---|---|
| M. Kunneman, et al. 2022 [30] | Adults with a clinical diagnosis of T2DM with a recent poorly controlled HbA1c (N = 350). | The trial was conducted in primary care practices from health systems among T2DM patient with a recent (<12 months) HbA1c >7.3% in Minnesota, Wisconsin), USA. | Diabetes medication choice conversation aid by patients and clinicians during the clinical encounter and patients could take home a one-page handout version of the conversation aid. Clinicians received training on how to use the conversation aid during a 10 min group session. | Face to face at point of care. | I = 8.9 (1.4) C = 8.9 (1.2) | 12 | Significant HbA1c level difference was not observed between-arms and future interventions may need to focus specifically on patients with signs of poor treatment fit. |
| A. Wollny, et al. 2019 [39] | A T2DM with poorly controlled HbA1c, 3 months before the study (N = 720). | DEBATE trial was conducted in primary care among patients with poorly controlled HbA1c ≥ 8.0% in the 3 months before the study in Germany. | Patient-centered communication and SDM among GPs and their patients with poorly controlled T2DM to reduce the level of HbA1c. | Face to face communication at point of care. | I = 8.39 (1.40) C = 8.99 (3) | 24 | The decline of the HbA1 level from base line to endpoint was statistically significant in both intervention and control group. However, there was no statistically significant difference between the groups. |
| M. T. Vo, et al. 2019 [38] | Patients diagnosed with T2DM and with most recently HbA1c ≥8.0% (N = 712). | A pragmatic, provider-randomized, multi-site clinical trial in primary care practices within Kaiser Permanente Northern California in USA. | Patients received a secure electronic message from their primary providers asking them to prepare for their visit by reviewing important areas of care and identifying their top priorities for discussion at a scheduled visit to make shared decision. | Mixed approaches: area of care for discussion was identified via email and the discussion for SDM at point of care. | I = 9.30 (1.2) C = 9.3 (1.3) | 12 | Email-based pre-visit intervention resulted in improved measures of visit interaction for SDM but did not significantly improve glycaemic control relative to usual care. |
| J. C. Lauffenburger, et al. 2019 [41] | T2DM patients >18 years of age who had evidence of poor diabetes control. (N = 1362). | The trial was conducted in Horizon Blue Cross Blue Shield among T2DM who filled 1 or more oral hypoglycaemic agents within the 12 months prior to randomization with ≥ HbA1c 8% in New Jersey, USA. | The intervention was delivered over the telephone by a clinical pharmacist and consisted of a 2-step process that integrated brief negotiated interviewing and SDM to identify patient goals and options for enhancing diabetes management. | Online telephone based SDM to identify patient goals and options. | I = 9.30 (1.6) C = 9.40 (1.6) | 12 | A novel telephone-based patient-centered intervention did not improve HbA1c in patients with poorly controlled diabetes compared with usual care. |
| H. Den Ouden, et al, 2017 [27] | T2DM patients age of 60–80 years known with diabetes for 8–12 years. (N = 141). | OPTIMAL trial was conducted in primary care practices among T2DM patients in Dutch. | OPTIMAL decision support aid which simple paper-based tool and the intervention included SDM with personalized goal setting and the use of a decision aid. | Face to face at point of care using decision support aid. | I = 6.8(3.0) C = 6.9 (3.0) | 24 | Significant improvement in HbA1 level was not observed between the intervention and control group that could be due to low baseline levels of HbA1 among participants. |

(*Continued*)

**Table 1.** (Continued)

| Study | Study population | Study settings | Interventions | Mode of intervention (SDM) delivery | HbA1 level (%) baseline, mean (SD) | Follow-up period in month | Main conclusions |
|---|---|---|---|---|---|---|---|
| M. O'Donnell 2016 [32] | T2DM patients for ≥1 year, aged 18–75 years, (N = 94). | Pragmatic pilot randomized controlled trial was conducted in outpatient diabetes clinic in a university hospital in the West of Ireland. | Sharing personalized clinical information with people T2DM prior to their out-patient consultation on patient involvement using SDM booklets as decision aid. | Face to face at outpatient diabetes clinics. | I = 7.6 (3.5) C = 7.8 (3.6) | 12 | No significant difference was observed in glycaemic level. Differences in HbA1c might have been demonstrated if sub-optimal glycaemic control was used as an inclusion criterion. |
| R. J. Mullan, et al. 2009 [34] | T2DM patients for at ≥ 1year who had a scheduled appointment with clinician. (N = 85). | The trial was conducted in primary care and family medicine sites within the Mayo Clinic Health System and Olmsted Medical Center, southeast Minnesota, USA. | Diabetes Medication Choice, a decision aid that described 5 antihyperglycaemic drugs, their treatment burden (adverse effects, administration, and self-monitoring demands), and impact on HbA1c levels. | Face to face during the clinical encounter | I = 7.47 (0.58) C = 7.63 (0.65) | 6 | The Diabetes Medication Choice cards were helpful to patients and clinicians and improved patient involvement in making decisions about diabetes medications and slightly improved the HbA1c level. |
| S. Buhse, et al. 2015 [26] | T2DM patients without diagnosis of ischaemic heart disease or stroke. (N = 143) | The trial was conducted in single-center diabetes clinic providing care according to the national disease management programme in Germany. | The ISDM-P trial was executed by diabetes educators. Core component was a patient decision aid on the prevention of myocardial infarction supplemented by a 90 min group teaching session. | Face to face during the clinical encounter. | I = 6.9(0.7) C = 7.2 (0.7) | 6 | More patients in the SDM group achieved their HbA1c goals, since they had set slightly higher HbA1c goals after the teaching, however, there was no significand difference of HbA1c level in both groups. |
| N. Mathers, et al. 2012 [31] | T2DM patients who were taking at least 2 oral glucose and HbA1c level > 7.4%. (N = 167). | Pragmatic trial of PANDAs decision aid in people with T2DM at general practices in Sheffield, Rotherham and Doncaster, UK. | The intervention comprised three components: PDA; healthcare professional training workshop and use of the PDA in a consultation. | Face to face during the clinical encounter | I = 8.60 (1.9) C = 8.80 (0.98) | 12 | The use of the PANDAs decision aid by healthcare professionals' usual clinical practice with T2DM patients who were making treatment choices in general practice had no demonstrable effect on glycaemic control. |
| S. Greenfield, et al. 1998 [37] | T2DM patients >18 years but <75 years of age. (N = 59) | The trial was con ducted in two university hospital-outpatient clinics among T2DM patients in USA. | An intervention was designed to increase the involvement of patients in medical decision facilitating participation of patients in medical care to make shared decision on their medical care and physician-patient conversations were audiotaped. | Face to face during the clinical encounter | I = 10.56 (2.1) C = 10.26 (1.96) | 3 | Patients in intervention group reported significant improvement in glycaemic level compared the patient control group. |

(*Continued*)

**Table 1.** (Continued)

| Study | Study population | Study settings | Interventions | Mode of intervention (SDM) delivery | HbA1 level (%) baseline, mean (SD) | Follow-up period in month | Main conclusions |
|---|---|---|---|---|---|---|---|
| K. M. Rost, et al. 1991 [29] | T2DM patients who had HbA1c. level >8% (N = 51) | The trial was undertaken in the Clinical Research Center of Washington University, USA. | Enhancement of patient information seeking and decision-making during hospitalization to metabolic control and functional status in patients T2DM by providing 45-min patient activation intervention and a 1-h self-administered booster. | Face to face during the clinical encounter | I = 13.1 (3.5) C = 13.6 (3.6) | 4 | Significant improvement in metabolic control was observed among patients in experimental group compared patients in control group. |
| Karagiannis T, et al. 2016 [28] | Adults who have had T2DM for ≥1 year. (N = 204) | The trial was undertaken in primary and secondary care practices in Greece. | Clinicians and patients used a decision aid when choosing among antihyperglycaemic medications. | Face to face during the clinical encounter | I = 8.4(0.8) C = 8.5 (0.8) | 6 | No significant HbA1c differences was found across arms. |

*Notes*: For all studies the comparison was usual care, I: Intervention, C: Control

clinical encounter at point of care. In 2 studies [35, 41], the mode of SDM was online (web-based and telephone-based).

## Effectiveness of shared decision-making for glycaemic control

**Qualitative evidence.** Glycated hemoglobin (HbA1c%), the gold standard indicator for glycaemic control was used in the current systematic review and meta-analysis to evaluate the effectiveness of SDM compared to usual diabetes care. Worldwide, extensive clinical studies on the complications of diabetes have significantly enhanced the value of HbA1c as a glycaemic control indicator [42]. As a result, even though the improvement was statistically insignificant, some of the studies indicated that patients who engaged in shared decision-making had improved HbA1c levels when compared to those who received usual diabetes care [25, 29, 33–35, 37, 40].

Patients in the intervention group who engaged in SDM had a substantial reduction in their HbA1c level from baseline to the three-month follow-up visit as compared to patients in the control group who received usual care. Patients in the ISDM group achieved their HbA1c targets more successfully than those in the control group, even though there was no statistically significant difference in HbA1c levels between the groups [25, 26]. An SDM intervention for individuals with type 2 diabetes appears to be feasible, and outcomes in primary care showed a trend toward improvement in HbA1c level compared to patients who received usual care [20].

Significant improvement in metabolic control was observed among patients in the experimental group participating in SDM compared to patients in the control group receiving usual care [29]. Between the baseline and the three-month follow-up visit, the HbA1c level was significantly lower in the patients who received the SDM program than in the patients in the control group who received usual care [33].

The diabetic medication choice cards promoted patient involvement in choosing diabetic drugs, helped patients and clinicians make decisions, and slightly decreased HbA1c levels [34]. The intervention program with SDM improved HbA1c levels compared to the patient who received usual care [35] and, compared to the patient who received usual care, patients in the

intervention group who participated in SDM reported a substantial improvement in their glycaemic level [37].

The intervention with the SDM program, which used patient-driven goals, was helpful to patients and clinicians. It significantly lowered HbA1c among patients in the intervention group compared to patients in the control group [40]. On the other hand, the review study found no evidence of a significant difference between patients who participated in decision-making using various decision aids and those who received usual care regarding the level of glycated hemoglobin reduction or the achievement of the targeted goal of glycated hemoglobin level [27, 28, 30–32, 36, 38, 39, 41].

**Quantitative evidence.** The quantitative evidence from a meta-analysis of the current review estimated the pooled mean effect of SDM on reduction of glycated hemoglobin for glycaemic control in T2DM adult patients compared to usual care. The data was extracted from 17 RCT studies [25–41], analyzed, and presented in a forest plot (Fig 4).

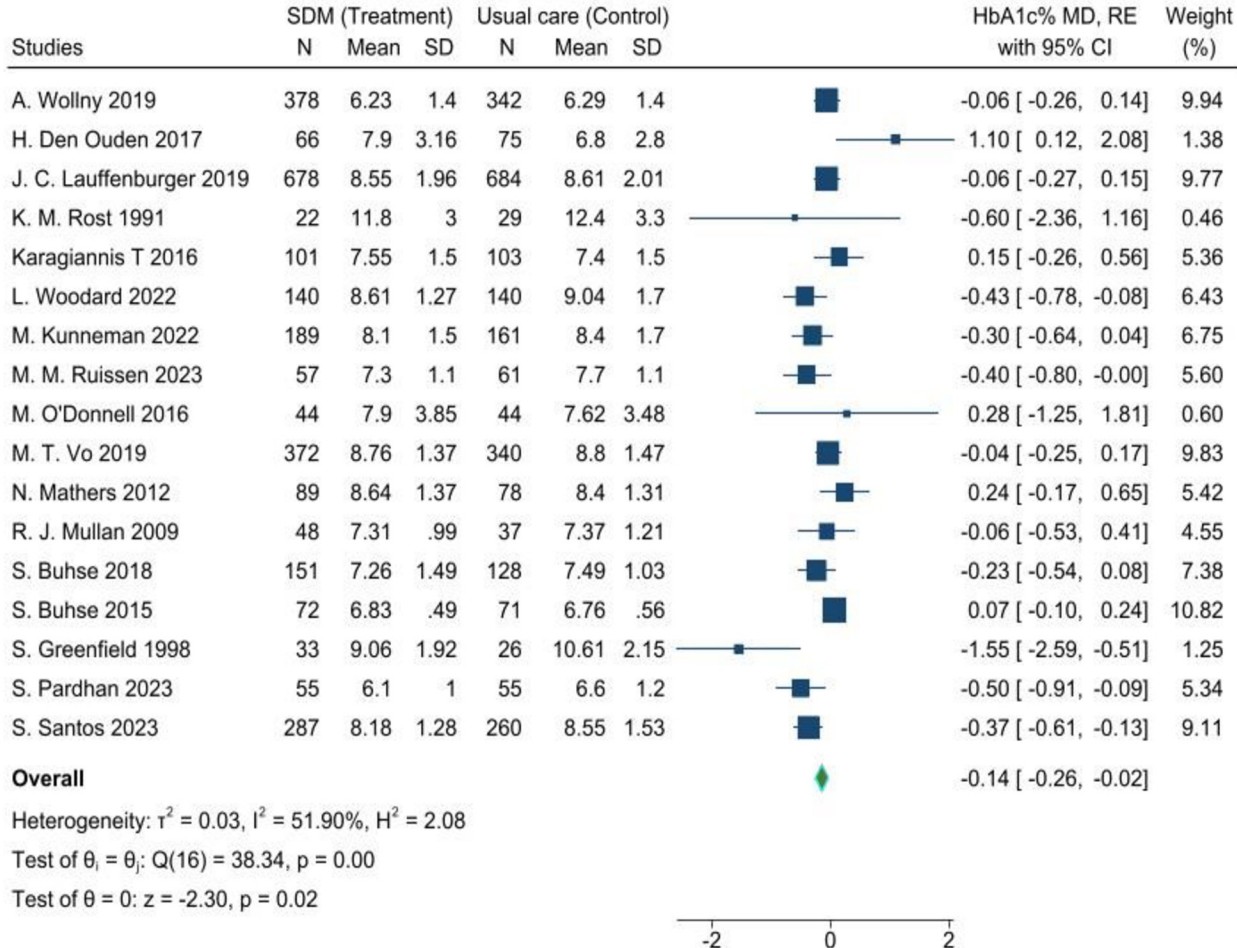

**Fig 4. Comparison of the effectiveness of shared decision-making vs. usual diabetes care for glycaemic control among T2DM adult patients, 2024.** Thus, the pooled mean differences in HbA1c% level were estimated using a random-effects model (REM). As a result, the estimated overall effect showed that the shared decision-making significantly lowered HbA1c by 0.14% compared to usual care in adult patients diagnosed with T2DM, (95% CI = [-0.26, -0.02], P = 0.02).

## Subgroup analysis

Subgroup analysis was performed for the mode of intervention delivery, follow-up period, and level of HbA1c% at baseline to estimate the HbA1c% mean differences between patients who participated in SDM and those who received usual care. Consequently, the SDM significantly decreased the level of HbA1c by 0.14%, (95% CI = [-0.28, -0.01], P = 0.00) when shared decisions were made in person or face-to-face at the point of care, but among patients who engaged in online (web-based or telephone-based) shared decision-making with their healthcare providers, there was no statistically significant reduction in HbA1c levels (Fig 5).

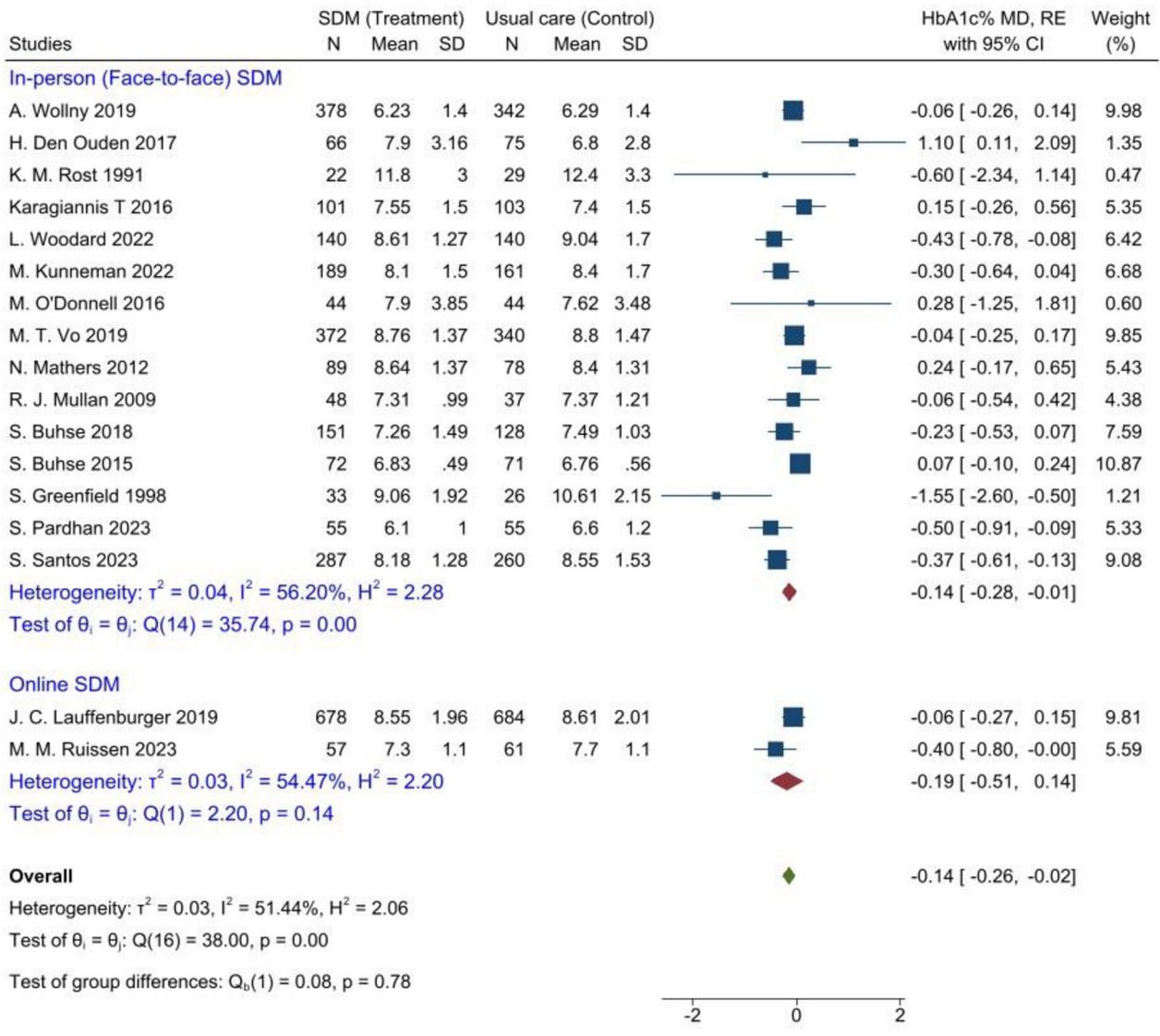

**Fig 5. Subgroup analysis for mode of intervention to evaluate the effectiveness of SDM for glycaemic control compared to usual care, 2024.** Subgroup analysis was also carried out, and the length of the intervention (follow-up period) was taken into consideration. The follow-up duration was a minimum of three months and a maximum of twenty-four months; the follow-up period's median was 12 months. As a result, the studies in which the SDM intervention was followed up for <12 months significantly reduced HbA1c% by 0.24%, (95% CI = [-0.45, -0.03], P = 0.00). However, the significant reduction in HbA1c% was not observed in group of studies in which the SDM intervention was followed-up for ≥ 12 months (Fig 6).

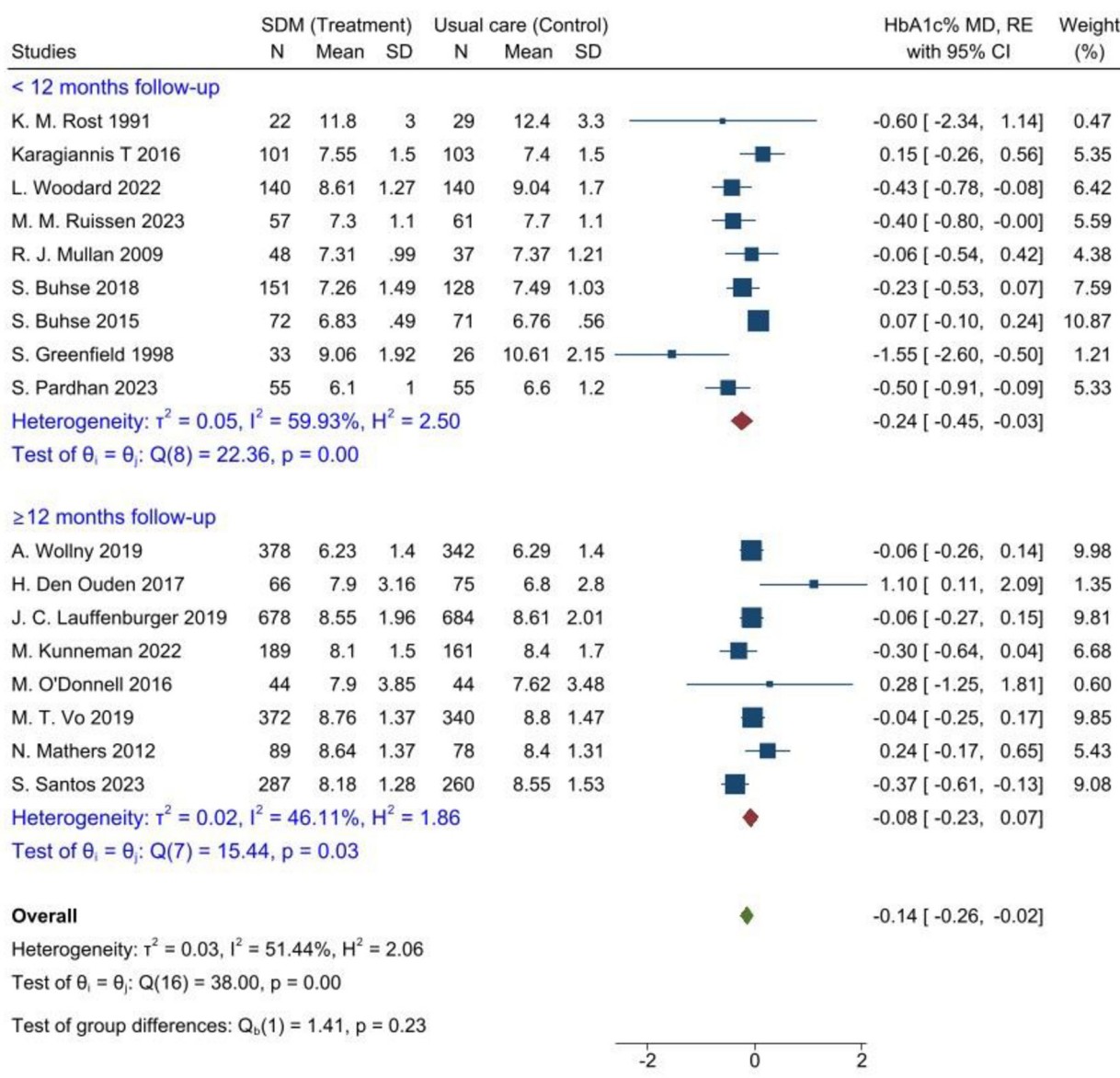

**Fig 6. Subgroup an analysis for intervention follow-up period to evaluate the effectiveness of SDM for glycaemic control compared to usual care, 2024.** Also, subgroup analysis was performed on the category of the studies based on the baseline HbA1c% level, which showed the studies that included T2DM patients with poorly controlled glycaemic levels (HbA1c ≥ 8%) and patients with well controlled glycaemic levels (HbA1c< 8%) (Fig 7).

## Sensitivity analysis

The sensitivity analysis was performed using a leave-one-out meta-analysis to quantify the impact of each study on the overall effect size of the mean difference in HbA1c% in T2DM adult patients (Fig 8).

## Certainty of the evidence

The overall certainty of the evidence in the current meta-analysis was assessed. Accordingly, the overall certainty of the evidence generated by the current meta-analysis was graded as

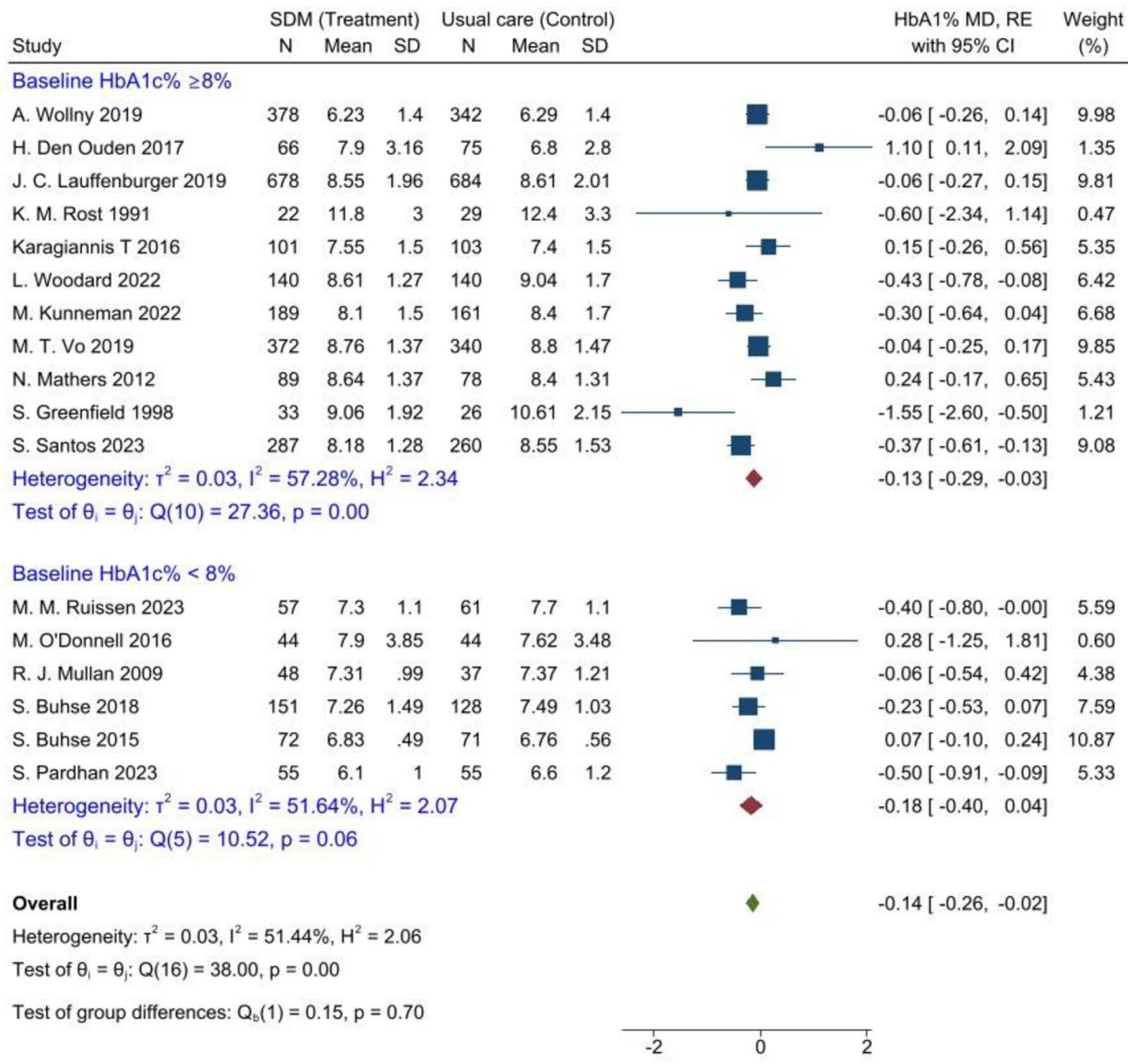

**Fig 7. Subgroup analysis for HbA1c% baseline level to evaluate the effectiveness of SDM for glycaemic control compared to usual care, 2024.** As a result, in T2DM patients with poorly controlled glycaemic level, shared decision making significantly reduced level of HbA1c by 0.13%, (95% CI = [-0.29, -0.03], P = 0.00). Nevertheless, significant reduction in HbA1c was not observed in T2DM patients with < 8% HbA1c level participating in SDM compared to patients who received usual care.

moderate, which indicated that using a 95% CI; shared decision-making likely resulted in a slight reduction in the level of glycated hemoglobin, and further study is likely to have an important impact on our confidence in the estimate of effect and may change the current estimates (Table 2).

## Discussion

Glycated hemoglobin (HbA1c) reflects average blood sugar levels over the past 2 to 3 months, and it is a crucial marker for diabetes management [43]. Since HbA1c is a reliable indicator of

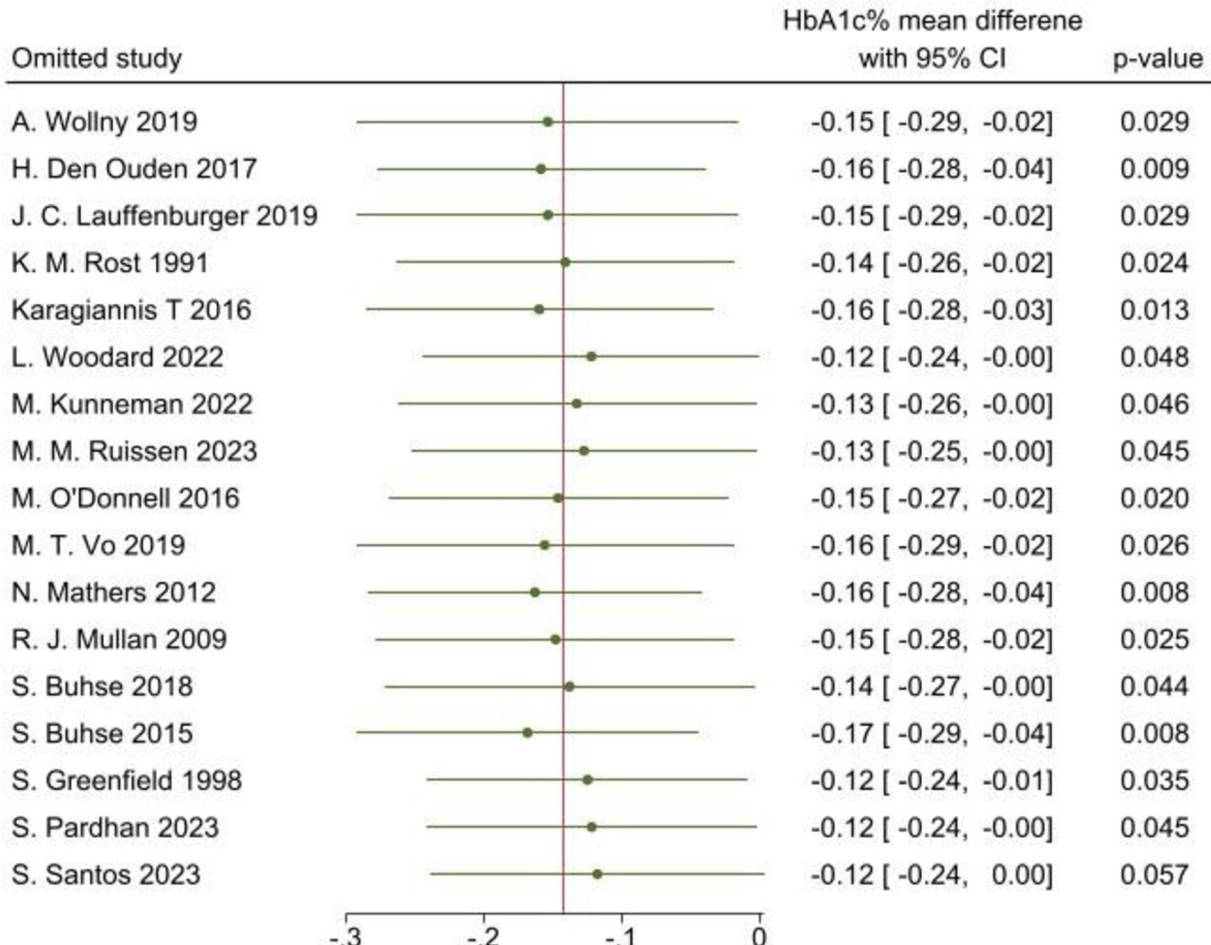

**Fig 8. Sensitivity analysis using leave-one-out meta-analysis to evaluate the effectiveness of SDM for glycaemic control among adult T2DM patients compared to usual care, 2024.**

both lipid profile and long-term glycaemic control, monitoring glycaemic control with HbA1c may also help to identify diabetes patients who are more likely to experience cardiovascular conditions [44].

The current meta-analysis showed that when diabetes patients participated in shared decision-making, a slight decrease in HbA1c levels was observed, but a statistically significant reduction compared to those patients who used usual care, which could lead to improved long-term health outcomes. For patients with diabetes, this small reduction in glycated hemoglobin may reduce the risk of complications related to the disease and contribute to better overall health. For instance, a report from the UK showed that each 1% decrease in HbA1c was associated with significant reductions of 21% in diabetes-related deaths, 14% in myocardial infarction, and 37% in microvascular complications [45].

Even though there were some discrepancies between studies, the overall effect of the SDM was a significantly reduced level of glycated hemoglobin among patients who participated in their diabetes-related health care decisions. This finding is in line with different studies that revealed when health care is patient-centered and engages patients in decision-making, it

**Table 2. Summary of findings of GRADE certainty evidence of the meta-analysis for the effectiveness of shared decision-making for glycaemic control compared to usual care among type 2 diabetes mellitus adult patients, 2024.**

Shared decision-making compared to Usual care for controlling HbA1c in T2DM adult patients

**Patient or population:** T2DM adult patients
**Question**: Does SDM significantly reduce HbA1c compared to usual care among T2DM patients?
**Study design:** Randomized controlled trials (RCTs)
**Intervention:** Shared decision making, (N = 2782)
**Comparison:** Usual care, (N = 2634)

| Outcomes | Anticipated absolute effects* (95% CI) | | Relative effect (95% CI) | № of participants (studies) | Certainty of the evidence (GRADE) | Comments |
|---|---|---|---|---|---|---|
| | Risk with Usual care | Risk with Shared decision making | | | | |
| Level of glycated hemoglobin (HbA1c) assessed with: % follow-up: median 12 months | The mean level of glycated hemoglobin was 8.12% | MD 0.14% lower (0.26 lower to 0.02 lower) | - | 5416 (17 RCTs) | *Moderate* [a] | Shared decision-making likely results in a slight reduction in level of glycated hemoglobin. |

*Notes*:

*The risk in the intervention group and its 95% confidence interval is based on the assumed risk in the comparison group and the relative effect of the intervention and its 95% CI.

CI: Confidence interval.

MD: Mean difference.

⊕⊕⊕◯

[a]: Certainty of the evidence was moderate using GRADE.

enables patients to change their glycated hemoglobin level over time [46]. T2DM patients with uncontrolled glycaemic levels who participated in shared medical appointments were able to significantly control their glycaemic level by reducing glycated hemoglobin compared to those patients who did not participate in shared medical appointments [47], and patient-centered care in diabetes that facilitates shared decision-making was both clinical and cost-effective among the patients with baseline HbA1c >8.5% [48].

Another similar study showed that, when integrated personalized diabetes management is provided, engaging patients in the decision-making process, there is a significant improvement in glycaemic control compared to the patients with usual diabetes management and care [49], and when patients with poorly controlled glycaemic levels participated in SDM, they were able to reduce the level of glycated hemoglobin significantly.

Current review evidence indicated that patients with poorly controlled glycaemic levels who participated in goal-setting through shared decision-making significantly improved their glycaemic level. Likewise, various studies support this finding, demonstrating that patients with higher initial glycaemic levels who engaged in shared decision-making experienced a reduction in HbA1c level [49, 50]. Similarly, in diabetes patients, poorly controlled glycated hemoglobin levels can serve as an indicator of the substantial effectiveness of healthcare interventions for glycaemic control [51]. This may be because when the baseline level of glycated hemoglobin is already well-managed and controlled, any improvement due to the intervention of SDM might go unnoticed.

Patients who felt empowered made shared decisions about their diabetes care and treatment options, and they also exhibited confidence in their diabetes knowledge. They were able to maintain their glycaemic level stable even while the disease progressed. Therefore, it is crucial to empower patients to participate in shared decision-making and long-term follow-up to maintain glycaemic levels and control secondary complications associated with diabetes [52].

Results from other studies showed that diabetes patients engaging in structured goal setting significantly lowered their glycated hemoglobin levels [53]. The patient empowerment

approach was successful in achieving optimal metabolic control, allowing patients to improve their health status [54], and compared to physician decision-making, the shared decision-making on glycated hemoglobin level showed that the SDM had led to better glycaemic control in diabetes patients [6].

There were discrepancies between these studies, despite the fact that the diabetes patients participating in shared decision-making showed a significant reduction in glycated hemoglobin levels when compared to the patient group receiving diabetes usual care. As a result, about one-third of the trials [33, 35–37, 40] demonstrated the favorable effects of shared decision-making, indicating that it was a significantly effective approach for lowering glycated hemoglobin levels in diabetic patients.

On the other hand, when compared to usual diabetes care, two-third of the trials [25–32, 34, 38, 39, 41] demonstrated that shared decision-making did not significantly lower the level of glycated hemoglobin in people with diabetes. These discrepancies may result from differences in patient characteristics, professional characteristics, patient-provider relationships, the amount of time spent in collaborative decision-making, and information provision-related factors [55].

Furthermore, the fact that all the study applied various forms of decision aids and in different settings that may have had a negative impact on how well adult patients with type 2 diabetes were able to maintain glycaemic control through shared decision-making processes.

## Limitations

Despite the fact that the current review study well evaluated the effectiveness of shared decision-making among T2DM adult patients, the study could have some limitations; the effectiveness of shared decision-making might be influenced by the level of patient empowerment and health literacy that were not considered in the current review. In addition, the current review included all studies that included all patients with controlled and uncontrolled glycaemic levels that could affect the pooled effect size of glycated hemoglobin levels. The literature search for the current review was limited to only MEDLINE, PubMed, Cochrane library, HINARI and Google Scholar. On the other hand, the review may not have included the literature that were only indexed in other databases.

## Conclusions

In conclusion, the results of the current systematic review and meta-analysis indicated that shared decision-making significantly reduced the glycated hemoglobin levels in comparison to usual diabetes care and it was the effective approach for the control of glycaemic levels in type 2 diabetes adult patients that could lead to improved long-term health outcomes, reducing the risk of diabetes-related complications. Therefore, we strongly suggest that health care providers and policy-makers should integrate shared decision-making into diabetes health care and management. The further study should focus on the level of patients' empowerment, health literacy and standardization of decision supporting tools to evaluate the effectiveness of shared decision-making in diabetes patients.

## Supporting information

**S1 File. Electronic search strategy.**
(DOCX)

**S1 Dataset.**
(XLSX)

## Author Contributions

**Conceptualization:** Edosa Tesfaye Geta, Dufera Rikitu Terefa, Wase Benti Hailu.

**Data curation:** Edosa Tesfaye Geta, Dufera Rikitu Terefa, Wase Benti Hailu, Emiru Merdassa.

**Formal analysis:** Edosa Tesfaye Geta, Dereje Chala Diriba.

**Funding acquisition:** Edosa Tesfaye Geta.

**Investigation:** Edosa Tesfaye Geta.

**Methodology:** Edosa Tesfaye Geta, Dufera Rikitu Terefa, Wase Benti Hailu, Wolkite Olani, Emiru Merdassa, Markos Dessalegn, Miesa Gelchu, Dereje Chala Diriba.

**Project administration:** Edosa Tesfaye Geta.

**Software:** Edosa Tesfaye Geta, Emiru Merdassa.

**Supervision:** Edosa Tesfaye Geta.

**Validation:** Edosa Tesfaye Geta, Dufera Rikitu Terefa, Wase Benti Hailu, Wolkite Olani.

**Visualization:** Edosa Tesfaye Geta, Wase Benti Hailu.

**Writing – original draft:** Edosa Tesfaye Geta, Wolkite Olani, Markos Dessalegn, Miesa Gelchu, Dereje Chala Diriba.

**Writing – review & editing:** Edosa Tesfaye Geta, Markos Dessalegn, Miesa Gelchu, Dereje Chala Diriba.

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
