## [Decision Letter · Decision Letter 0]

24 Mar 2024

PONE-D-24-07517Effectiveness of shared decision-making for glycaemic control among type 2 diabetes mellitus adult patients: A Systematic review and Meta-analysisPLOS ONE

Dear Dr. Geta,

Thank you for submitting your manuscript to PLOS ONE. After careful consideration, we feel that it has merit but does not fully meet PLOS ONE’s publication criteria as it currently stands. Therefore, we invite you to submit a revised version of the manuscript that addresses the points raised during the review process.

We look forward to receiving your revised manuscript.

Kind regards,

Chikezie Hart Onwukwe

Academic Editor

PLOS ONE

Journal Requirements:

2. In the online submission form, you indicated that [The data extracted from included studies and analyzed in this review are available from the corresponding author based on the reasonable request.]. 

Reviewers' comments:

Reviewer's Responses to Questions

**Comments to the Author**

1. Is the manuscript technically sound, and do the data support the conclusions?

Reviewer #1: Partly

Reviewer #2: Partly

2. Has the statistical analysis been performed appropriately and rigorously? 

Reviewer #1: I Don't Know

Reviewer #2: Yes

3. Have the authors made all data underlying the findings in their manuscript fully available?

Reviewer #1: Yes

Reviewer #2: Yes

4. Is the manuscript presented in an intelligible fashion and written in standard English?

Reviewer #1: No

Reviewer #2: No

5. Review Comments to the Author

Reviewer #1: Thank you for inviting me to be the reviewer of this manuscript “Effectiveness of shared decision-making for glycaemic control among type 2 diabetes mellitus adult patients: A Systematic review and Meta-analysis”.

Edosa T et al. has submitted a Systematic review and Meta analysis of literature on Shared decision-making for glycaemic control among type 2 diabetes mellitus adult patients. Given the increasing burden of Diabetes and diabetes related complications, the chosen topic gains significance.

I am giving my observations and comments for the author to consider:

Introduction:

Line 62,63 - Please be consistent in use of abbrevations –HbA1c or HbA1C

Line 64 -The statement given here seems irrelevant and the reference doesn’t match

Line 66,67 - error in the statement. It should be HbA1c ≤ 7%

Line 76 – the word "current" may be removed as the ADA guideline cited here was 2015 guidelines

Line 96 – repetitive word”in”

Overall, the statements under introduction seem repetitive and the flow of language is not very professional and uncomfortable and I would suggest making it more concise.

Methodology:

This systematic review is observed to have followed rigorous methodology, have applied relevant search strategies and analysis methods.

The authors have provided an approved protocol (PROSPERO) along with the methods employed for article inclusion.

The manuscript incorporates detailed descriptions of the search methodology, study selection, and data extraction procedures. PRISMA checklist, ROB for quality of included studies, Heterogeneity tests has been performed and shared.

Methods:

The types of studies included in this systematic review are thoroughly described. The PRISMA flow diagram is easy to follow and complete. ROB for the quality of studies was performed and the details are shared

Line 177 – typo error

Discussion and Conclusion:

Though the strength and limitations of the study are well thought and documented, the conclusion driven out of this study ( line 349-351) did not match the discussion points (line 330-332). I suggest to rewrite the discussion in a more intelligent way and draw precise conclusions

Reviewer #2: Dear Authors,

The manuscript addresses a significant subject due to the progressive growth of the population with diabetes and its complications. It also emphasizes the importance of evaluating the shared decision-making for glycaemic control effectiveness, as it is widely recommended but not primarily followed in most places, and identifying what may influence its outcomes. However, this manuscript has some points that need to be clarified.

#1 - on page 3, line 66, it says ...'(HbA1c) ≥7% (53 mmol/mol) is considered a favorable indicator of glycemic control and is the recommended treatment a goal...'

# 2 - on page 5, lines 111-113, it says ...'the current systematic review and meta-analysis aimed to evaluate the effectiveness of shared decision-making compared to usual diabetes care for glycaemic control among type 2 diabetes mellitus adult patients.' Nevertheless, in the item 'Study selections and eligibility' on page 6, when you describe the 'C' of PICO criteria, it is not expressed that you will compare a group under usual care and another with SDM.

#3 - Related to observation #2, I wonder why the authors included in the review the study of Corser et al. (reference 26), which has only an intervention group evaluation and no control group with usual care. Moreover, this may interfere with the whole evaluation.

#4 - on page 9, Table 1, it says 18 RCTs, when it is 17; in the item Study events rate, there are no percentage results, and in the risk with usual care, the authors present the mean effect of SDM on HbA1C.

#5 - the whole manuscript deserves a writing review.

6. PLOS authors have the option to publish the peer review history of their article (what does this mean?). If published, this will include your full peer review and any attached files.

Reviewer #1: No

Reviewer #2: No

---

## [Author Response · Author response to Decision Letter 0]

10 Apr 2024

Responses to reviewers’ comments

Manuscript Title: Effectiveness of shared decision-making for glycaemic control among type 2 diabetes mellitus adult patients: A Systematic review and Meta-analysis

Dear Editor, 

Thank you for your suggested revision that has helped us to improve the manuscript. We have undertaken revision and modified the manuscript. We attached, you will find our revised and modified manuscript and we explained the substance modification and below you will find responses to each comment.

Reviewer 1 Evaluation

Dear reviewer 1, thank you for understanding our research work, your comments and suggestions. We attached, you will find our revised and modified manuscript and we explained the substance modification and below you will find responses to each comment.

Introduction:

1. Line 62,63 - Please be consistent in use of abbreviations –HbA1c or HbA1C

Response 1: Thank you, we have corrected it to HbA1c.

2. Line 64 -The statement given here seems irrelevant and the reference doesn’t match.

Response 2: Thank you, we checked it, and the irrelevant statement has been removed.

3. Line 66,67 - error in the statement. It should be HbA1c ≤ 7%.

Response 3: Now it has been Corrected.

4. Line 76 – the word "current" may be removed as the ADA guideline cited here was 2015 guidelines.

Response 4: Thank you, we have corrected.

5. Line 96 – repetitive word”in”

Response 5: we checked and corrected it.

6. Overall, the statements under introduction seem repetitive and the flow of language is not very professional and uncomfortable, and I would suggest making it more concise.

Response 6: Thank you. We have made it more concise through revision.

Methodology:

7. This systematic review is observed to have followed rigorous methodology, have applied relevant search strategies and analysis methods. The authors have provided an approved protocol (PROSPERO) along with the methods employed for article inclusion. The manuscript incorporates detailed descriptions of the search methodology, study selection, and data extraction procedures. PRISMA checklist, ROB for quality of included studies, Heterogeneity tests has been performed and shared.

Response 7: Thank you for your understanding of our research work.

Methods:

8. The types of studies included in this systematic review are thoroughly described. The PRISMA flow diagram is easy to follow and complete. ROB for the quality of studies was performed and the details are shared.

9. Line 177 – typo error

Response 9: Thank you, we corrected it.

Discussion and Conclusion:

10. Though the strength and limitations of the study are well thought and documented, the conclusion driven out of this study ( line 349-351) did not match the discussion points (line 330-332). I suggest to rewrite the discussion in a more intelligent way and draw precise conclusions.

Response 11: Thank you. We revised the discussion and conclusion parts.

 

Reviewer 2 Evaluation

Dear reviewer 1, thank you for understanding our research work, your comments and suggestions. We attached, you will find our revised and modified manuscript and we explained the substance modification and below you will find responses to each comment.

 The manuscript addresses a significant subject due to the progressive growth of the population with diabetes and its complications. It also emphasizes the importance of evaluating the shared decision-making for glycaemic control effectiveness, as it is widely recommended but not primarily followed in most places, and identifying what may influence its outcomes. However, this manuscript has some points that need to be clarified.

1. on page 3, line 66, it says ...'(HbA1c) ≥7% (53 mmol/mol) is considered a favorable indicator of glycemic control and is the recommended treatment a goal...'

Response 1: Thank you; it was an error, and now it has been corrected to HbA1c ≤7% (53 mmol/mol).

2. On page 5, lines 111-113, it says ...'the current systematic review and meta-analysis aimed to evaluate the effectiveness of shared decision-making compared to usual diabetes care for glycaemic control among type 2 diabetes mellitus adult patients.' Nevertheless, in the item 'Study selections and eligibility' on page 6, when you describe the 'C' of PICO criteria, it is not expressed that you will compare a group under usual care and another with SDM.

Response 2: Thank you, patients who received usual care of T2DM were comparison group (C ) and we have it made clear.

3. Related to observation #2, I wonder why the authors included in the review the study of Corser et al. (reference 26), which has only an intervention group evaluation and no control group with usual care. Moreover, this may interfere with the whole evaluation.

Response 3: Thank you. We excluded the study and performed a reanalysis; changes were made to the revised manuscript.

4. On page 9, Table 1, it says 18 RCTs, when it is 17; in the item Study events rate, there are no percentage results, and in the risk with usual care, the authors present the mean effect of SDM on HbA1C.

Response 4: Thank you. In our revised manuscript, we included 17 RCT studies, labeled Table 1 as Table 2, and moved the table to the end of the result section. The percentage results have been mentioned in the section on “participant characteristics,” and regarding the risk with usual care, we estimated the mean of HbA1c%, which was the primary outcome of the study, and the mean effect of SDM on HbA1c was estimated. The table has been updated and named Table 2, including details and clear information. To do so, we used a COCHRANE GRADE review to evaluate the certainty of the meta-analysis evidence from RCT studies.

5. The whole manuscript deserves a writing review.

Response 5: Thank you; we have revised the manuscript.

---

## [Decision Letter · Decision Letter 1]

29 Apr 2024

PONE-D-24-07517R1Effectiveness of shared decision-making for glycaemic control among type 2 diabetes mellitus adult patients: A Systematic review and Meta-analysisPLOS ONE

Dear Dr. Geta,

Thank you for submitting your manuscript to PLOS ONE. After careful consideration, we feel that it has merit but does not fully meet PLOS ONE’s publication criteria as it currently stands. Therefore, we invite you to submit a revised version of the manuscript that addresses the points raised during the review process.

We look forward to receiving your revised manuscript.

Kind regards,

Chikezie Hart Onwukwe

Academic Editor

PLOS ONE

Journal Requirements:

Reviewers' comments:

Reviewer's Responses to Questions

**Comments to the Author**

1. If the authors have adequately addressed your comments raised in a previous round of review and you feel that this manuscript is now acceptable for publication, you may indicate that here to bypass the “Comments to the Author” section, enter your conflict of interest statement in the “Confidential to Editor” section, and submit your "Accept" recommendation.

Reviewer #1: All comments have been addressed

Reviewer #2: All comments have been addressed

2. Is the manuscript technically sound, and do the data support the conclusions?

Reviewer #1: Yes

Reviewer #2: Yes

3. Has the statistical analysis been performed appropriately and rigorously? 

Reviewer #1: I Don't Know

Reviewer #2: Yes

4. Have the authors made all data underlying the findings in their manuscript fully available?

Reviewer #1: Yes

Reviewer #2: (No Response)

5. Is the manuscript presented in an intelligible fashion and written in standard English?

Reviewer #1: No

Reviewer #2: Yes

6. Review Comments to the Author

Reviewer #1: Thank you for inviting me to re-review the manuscript “Systematic review and Meta analysis of literature on Shared decision-making for glycaemic control among type 2 diabetes mellitus adult patients”. The authors have taken the suggestions and comments from the previous review and have included. The introduction is more precise and to the point. Changes suggested to the methodology had been included and the results are discussed accordingly.

PRISMA checklist on effectiveness of SDM included and very elaborate.

The limitations of the study are well thought of and listed.

There are still more typo errors which were overlooked. I would suggest to meticulously go through the manuscript and correct the error.Some are:

Line 39 - reduction

Line 62 – glycated

Line 80 – ‘between’ instead of ‘among’

Line 262,264 – HbA1c

Line 289 - reduction

Table 2 title – patients

Line 337 – review, participated

Reviewer #2: Dear Authors,

The authors answered all the reviewer's questions, and the included data helped better understand the study. However, there are still a few points to correct.

1. Fig 1 - PRISMA states 425 records, but on page 2, line 28, page 7, line 145, and page 9, line 193, it is mentioned as 445 records.

#2. On page 17, line 337, two words need to be corrected.

7. PLOS authors have the option to publish the peer review history of their article (what does this mean?). If published, this will include your full peer review and any attached files.

Reviewer #1: No

Reviewer #2: No

---

## [Author Response · Author response to Decision Letter 1]

1 May 2024

Manuscript title: Effectiveness of shared decision-making for glycaemic control among type 2 diabetes mellitus adult patients: A Systematic review and Meta-analysis

Manuscript number: PONE-D-24-07517R1

Response to Editor comments

Journal Requirements: Please review your reference list to ensure that it is complete and correct. If you have cited papers that have been retracted, please include the rationale for doing so in the manuscript text or remove these references and replace them with relevant current references. Any changes to the reference list should be mentioned in the rebuttal letter that accompanies your revised manuscript. If you need to cite a retracted article, indicate the article’s retracted status in the References list and also include a citation and full reference for the retraction notice.

Response: Dear editor, thank you for the comment and suggestion to revise our manuscript to be considered for publication. Since we did not include and cite the retracted paper in our current review, no changes have been made to the cited references. To be sure, we have reviewed the references and rechecked for retraction status. We found a correction to reference number 17, and the correction to this article was published on February 14, 2024. The update did not affect the results of our current systematic review and meta-analysis. In addition, we have included the DOI for references cited when available.

Response to reviewers’ comments

Reviewer 1

Thank you for inviting me to re-review the manuscript “Systematic review and Meta analysis of literature on Shared decision-making for glycaemic control among type 2 diabetes mellitus adult patients”. The authors have taken the suggestions and comments from the previous review and have included. The introduction is more precise and to the point. Changes suggested to the methodology had been included and the results are discussed accordingly. PRISMA checklist on effectiveness of SDM included and very elaborate. The limitations of the study are well thought of and listed.

There are still more typo errors which were overlooked. I would suggest to meticulously go through the manuscript and correct the error.

Dear Reviewer1: Thank you for reviewing our manuscript and providing comments to improve and revise it. We checked all comments and suggestions; changes were made to the revised manuscript.

Some are:

1. Line 39 – reduction

Response 1: Checked and corrected.

2. Line 62 – glycated

Response 2: Corrected.

3. Line 80 – ‘between’ instead of ‘among

Response 3: Corrected.

4. Line 262,264 – HbA1c

Response 4: Correction made.

5. Line 289 – reduction

Response 5: Correction made.

6. Table 2 title – patients

Response 6: Corrected.

7. Line 337 – review, participated

Response 7: Correction made.

Reviewer 2

The authors answered all the reviewer's questions, and the included data helped better understand the study. However, there are still a few points to correct.

Dear Reviewer 2: Thank you for reviewing our manuscript and providing comments to improve and revise it. We checked all comments and suggestions; changes were made to the revised manuscript.

1. Fig 1 - PRISMA states 425 records, but on page 2, line 28, page 7, line 145, and page 9, line 193, it is mentioned as 445 records.

Response 1: Corrections were made to 425 records.

2. On page 17, line 337, two words need to be corrected.

Response 2: We checked them, and corrections were made.

---

## [Decision Letter · Decision Letter 2]

14 May 2024

PONE-D-24-07517R2Effectiveness of shared decision-making for glycaemic control among type 2 diabetes mellitus adult patients: A Systematic review and Meta-analysisPLOS ONE

Dear Dr. Geta,

Thank you for submitting your manuscript to PLOS ONE. After careful consideration, we feel that it has merit but does not fully meet PLOS ONE’s publication criteria as it currently stands. Therefore, we invite you to submit a revised version of the manuscript that addresses the points raised during the review process.

We look forward to receiving your revised manuscript.

Kind regards,

Chikezie Hart Onwukwe

Academic Editor

PLOS ONE

Journal Requirements:

Reviewers' comments:

Reviewer's Responses to Questions

**Comments to the Author**

1. If the authors have adequately addressed your comments raised in a previous round of review and you feel that this manuscript is now acceptable for publication, you may indicate that here to bypass the “Comments to the Author” section, enter your conflict of interest statement in the “Confidential to Editor” section, and submit your "Accept" recommendation.

Reviewer #1: All comments have been addressed

Reviewer #2: All comments have been addressed

2. Is the manuscript technically sound, and do the data support the conclusions?

Reviewer #1: Yes

Reviewer #2: Yes

3. Has the statistical analysis been performed appropriately and rigorously? 

Reviewer #1: I Don't Know

Reviewer #2: Yes

4. Have the authors made all data underlying the findings in their manuscript fully available?

Reviewer #1: Yes

Reviewer #2: Yes

5. Is the manuscript presented in an intelligible fashion and written in standard English?

Reviewer #1: Yes

Reviewer #2: Yes

6. Review Comments to the Author

Reviewer #1: The authors have considered all the previous comments and has made changes accordingly except that there are still minor typographical errors which were overlooked:

Line 123 – and google scholar

Line 186 – analysis

Table 2 – typo patients

Line 339 - Typo review

344-346 – rewrite- "fair control"

With these changes incorporated, I recommend the manuscript is acceptable for publication

Reviewer #2: Dear Authors,

I am glad to re-review the manuscript "Effectiveness of shared decision-making for glycaemic control among type 2 diabetes mellitus adult patients: A Systematic Review and Meta-analysis," which addresses a subject that may contribute to improving diabetes outcomes. The authors accepted the reviewers' suggestions and have made the corrections.

7. PLOS authors have the option to publish the peer review history of their article (what does this mean?). If published, this will include your full peer review and any attached files.

Reviewer #1: No

Reviewer #2: No

---

## [Author Response · Author response to Decision Letter 2]

16 May 2024

Response to reviewers’ comments

Manuscript title: Effectiveness of shared decision-making for glycaemic control among type 2 diabetes mellitus adult patients: A Systematic review and Meta-analysis

Manuscript number: PONE-D-24-07517R2

Response to Editor comments

Journal Requirements: Please review your reference list to ensure that it is complete and correct. If you have cited papers that have been retracted, please include the rationale for doing so in the manuscript text or remove these references and replace them with relevant current references. Any changes to the reference list should be mentioned in the rebuttal letter that accompanies your revised manuscript. If you need to cite a retracted article, indicate the article’s retracted status in the References list and also include a citation and full reference for the retraction notice.

Response: Dear editor, thank you for the comment and suggestion to revise our manuscript to be considered for publication. Since we did not include and cite the retracted paper in our current review, no changes have been made to the cited references in the revised manuscript.

Response to reviewers’ comments

Reviewer 1:

Reviewer #1: The authors have considered all the previous comments and has made changes accordingly except that there are still minor typographical errors which were overlooked:

Dear reviewer1: Thank you for reviewing our revised manuscript and providing comments to improve and revise it to be considered for publication. We checked all comments and suggestions; changes were made to the revised manuscript.

1. Line 123 – and google scholar

Response 1: Thank you, checked and corrected 

2. Line 186 – analysis

Response 2 : A correction has been made.

3. Table 2 – typo patients

Response 3: A correction has been made.

4. Line 339 - Typo review

Response 4: A correction has been made.

5. 344-346 – rewrite- "fair control"

Response 5: Thank you, changes were made.

Reviewer 2:

Reviewer #2: Dear Authors,

I am glad to re-review the manuscript "Effectiveness of shared decision-making for glycaemic control among type 2 diabetes mellitus adult patients: A Systematic Review and Meta-analysis," which addresses a subject that may contribute to improving diabetes outcomes. The authors accepted the reviewers' suggestions and have made the corrections.

Dear reviewer 2: Thank you for reviewing our revised manuscript and considering it for publication.

---

## [Decision Letter · Decision Letter 3]

17 Jun 2024

Effectiveness of shared decision-making for glycaemic control among type 2 diabetes mellitus adult patients: A Systematic review and Meta-analysis

PONE-D-24-07517R3

Dear Dr.Edosa Tesfaye Geta,

We’re pleased to inform you that your manuscript has been judged scientifically suitable for publication and will be formally accepted for publication once it meets all outstanding technical requirements.

Kind regards,

Chikezie Hart Onwukwe

Academic Editor

PLOS ONE

Additional Editor Comments (optional):

Reviewers' comments:

Reviewer's Responses to Questions

**Comments to the Author**

1. If the authors have adequately addressed your comments raised in a previous round of review and you feel that this manuscript is now acceptable for publication, you may indicate that here to bypass the “Comments to the Author” section, enter your conflict of interest statement in the “Confidential to Editor” section, and submit your "Accept" recommendation.

Reviewer #1: All comments have been addressed

2. Is the manuscript technically sound, and do the data support the conclusions?

Reviewer #1: Yes

3. Has the statistical analysis been performed appropriately and rigorously? 

Reviewer #1: I Don't Know

4. Have the authors made all data underlying the findings in their manuscript fully available?

Reviewer #1: Yes

5. Is the manuscript presented in an intelligible fashion and written in standard English?

Reviewer #1: Yes

6. Review Comments to the Author

Reviewer #1: Thank you for inviting me to re-review the article. The authors have incorporated all the previous suggestions and edits in the manuscript. I don't have any further comment on the manuscript.

7. PLOS authors have the option to publish the peer review history of their article (what does this mean?). If published, this will include your full peer review and any attached files.

Reviewer #1: No

---

## [Editor Report · Acceptance letter]

25 Jun 2024

PONE-D-24-07517R3 

PLOS ONE

Dear Dr. Geta, 

I'm pleased to inform you that your manuscript has been deemed suitable for publication in PLOS ONE. Congratulations! Your manuscript is now being handed over to our production team.

Kind regards, 

on behalf of

Dr. Chikezie Hart Onwukwe 

Academic Editor

PLOS ONE